



# Climatology of aerosol properties at an atmospheric monitoring site on the Northern California coast

Erin K. Boedicker[1], Elisabeth Andrews[1], Patrick J. Sheridan[2], Patricia K. Quinn[3]

[1]Cooperative Institute for Research in Environmental Sciences (CIRES), University of Colorado, Boulder, Colorado
5   [2]Global Monitoring Laboratory, National Oceanic and Atmospheric Administration, Boulder, Colorado
[3]Pacific Marine Environmental Laboratory (PMEL), National Oceanic and Atmospheric Administration, Seattle, Washington

*Correspondence to*: Erin K. Boedicker (erin.boedicker@noaa.gov)

**Abstract**

10    Between April 2002 and June 2017, the National Oceanic and Atmospheric Administration (NOAA) Earth System

Research Laboratory (ESRL) made continuous measurements of a suite of in-situ aerosol optical properties at a

long-term monitoring site near Trinidad Head (THD), California. In addition to aerosol optical properties, between

2002 – 2006 a scanning humidograph system was operated and inorganic ion and total aerosol mass concentrations

were obtained from filter measurements. Combined analysis of these datasets demonstrates consistent patterns in

aerosol climatology and highlights changes in sources throughout the year. THD is predictably dominated by sea salt

aerosols, however, marine biogenic aerosols are the largest contributor to $PM_1$ in the warmer months. Additionally,

a persistent combustion source appears in the winter, likely a result of wintertime home heating. While the

influences of local anthropogenic sources from vehicular and marine traffic are visible in the optical aerosol data,

their influence is largely dictated by wind direction at the site. Comparison of the THD aerosol climatology to that

reported for other marine sites shows that the location is representative of clean marine measurements, even with the

periodic influence of anthropogenic sources.

## 1. Introduction

Aerosol particles affect the radiative balance both by scattering and absorbing solar radiation and by influencing

the properties of clouds (IPCC, 2013). In order to separate out the contribution of anthropogenic aerosol to aerosol

radiative forcing, the impact of natural aerosol particles must also be quantified (Andreae, 2007).  Because the

Earth's surface is dominated by oceans, marine aerosols are a dominant contributor to natural background aerosol

levels (Murphy et al., 1998; Jaeglé et al., 2011) although their contribution is variable in time and space. Marine



aerosols can be generated both through wave breaking – which generates sea salt aerosols (which consist of both

soluble ions and organic material)– and through gas-to-particle conversion of volatile organic compounds (VOCs)

generated by biogenic activity – leading to marine biogenic aerosols (O'Dowd and de Leeuw, 2007; Fitzgerald,

1991). The aerosol in coastal regions may be subject to significant impacts from local/regional anthropogenic

sources as well as long range transport, particularly in the northern hemisphere (Andreae, 2007); this can make it

difficult to definitively characterize the natural marine aerosol.  Understanding the marine aerosol is critical not just

for assessing direct aerosol forcing of a major natural aerosol type but also because marine aerosol plays a key role

in in cloud formation over much of the globe (Mayer et al., 2020; Hodshire et al., 2019; Carslaw et al., 2010).

Measurements over the open ocean (where anthropogenic influence is less pronounced) typically require mobile

platforms such as instrumented ships or aircraft, so remote coastal locations are often used as a surrogate for

sampling the marine atmosphere (O'Dowd et al., 2014; Bates et al., 1998; Brunke et al., 2004; Wood et al., 2015).

An advantage that surface-based coastal site measurements have over sampling from mobile platforms is that long-

term continuous information can be obtained allowing for the development of representative climatologies on

variable time scales (diurnal to annual) and the evaluation of long-term trends. Such measurements can then be used

to evaluate models, but whether the measurements are representative of a clean marine environment should also be

assessed (Wang et al., 2018; O'Dowd et al., 2014).

An established, long-term observatory can provide a platform for multiple instrument suites probing different

aspects of the atmosphere.  Long-term atmospheric monitoring deployments are (typically) limited in measurement

scope to robust and proven stand-alone techniques, in contrast to field campaign efforts, which frequently utilize

state-of-the-art and/or prototype instruments that only need operate for a short time and are attended by onsite

scientists.  Long-term datasets of atmospheric constituents provide an opportunity to describe inter-annual

variability and climatology on a variety of time scales, to identify trends and to develop a robust understanding of

relationships amongst the measured parameters which may not be possible from short-term studies. Analyses of

extended time series of atmospheric data can be used to provide context for field campaign observations (Brock et

al., 2011; Leaitch et al., 2020), assess model simulations (Spracklen et al., 2010; Browse et al., 2012), evaluate the

effectiveness of pollution control legislation and exposures (Murphy et al., 2008) and answer scientific questions

related to sources, atmospheric processes and the impacts of extreme events (Sorribas et al., 2015; Hallar et al.,

2015).



In April 2002, a surface monitoring site was established at Trinidad Head (THD), near the small town of Trinidad on the northern coast of California in order to study properties of atmospheric constituents (e.g., aerosol particles and ozone) entering the US prior to influence by North American sources. The start of measurements at

THD coincided with the 1-month intensive field campaign Intercontinental Transport and Chemical Transformation (ITCT 2K2) aimed at understanding how (relatively) short-lived species such as aerosol particles can be transported and detected far from their source and how these species change during transport (Parrish et al., 2004). While it was found that surface measurements at THD were not ideal for studying long-range transport (VanCuren et al., 2005), it was a reasonable location for making measurements of the clean marine environment. Thus, the site remained active

with a smaller suite of measurements as a long-term NOAA/GMD monitoring site for an additional 15 years, closing down in June 2017. This paper focuses on analysis of surface aerosol measurements observed at THD over the 2002-2017 period of measurements and addresses two main questions in the context of remote marine observations:

(1) What is the climatology (seasonal and diurnal cycles) of aerosol chemical and optical properties at THD?

(2) Are there any systematic relationships between aerosol chemical composition and aerosol optical properties?

The discussion of these results assesses implications in terms of aerosol sources and how well THD represents a clean marine site.

## 2.  Experimental Methods

The measurements described here were conducted at the atmospheric monitoring station on Trinidad Head,

California (41.054 N, 125.151 W, 107 m asl). The site is ~370 km (~230 miles) north of the San Francisco Bay area (population ~8.7 million) and ~350 km (~220 miles) south of Eugene, Oregon (population ~170,000) making it a location relatively remote from large local and regional sources of anthropogenic pollution. Aerosol chemical filters were collected over a four-year period from April 15, 2002 – May 2006 and aerosol optical measurements reported here cover a ~15-year period April 15, 2002 – June 1, 2017. Other measurements took place over more limited time

frames. **Table 1** describes the aerosol instruments operated at Trinidad Head (THD) and their periods of operation.



**Table 1**: Instrumentation at Trinidad Head

| Instrument | Make/ Model | Measurement | Time Period |
|---|---|---|---|
| **Condensation Nuclei Counter (CN)** | TSI 3760 | Aerosol number concentration (N) Size range: particle diameter > 14 nm | 4/2002 - 6/2017 |
| **Nephelometer*†** | TSI 3563 | Aerosol light scattering – total ($\sigma_{sp}$) and backwards ($\sigma_{bsp}$)- at three wavelengths (450, 550, and 700 nm) | 4/2002 - 6/2017 |
| **Particle Soot Absorption Photometer (PSAP)* †** | Radiance Research | Aerosol light absorption ($\sigma_{ap}$) at 565 nm (reported at 550 nm) | 4/2002-10/2005 |
| **Particle Soot Absorption Photometer (PSAP)* †** | Radiance Research | Aerosol light absorption ($\sigma_{ap}$) at three wavelengths (467, 530 and 660 nm) | 10/2005-6/2013 |
| **Continuous Light Absorption Photometer (CLAP)* †** | NOAA | Aerosol light absorption ($\sigma_{ap}$) at three wavelengths (467, 528 and 652 nm) | 12/2011-6/2017 |
| **Anemometer** | RM Young | Wind data (direction and speed) at 10 m | 9/2007-6/2017 |
| **Resistance temperature detector (RTD)** | Logan 4150 in aspirated enclosure | Ambient temperature at 2 and 10 m | 9/2007-06/2017 |
| **Relative humidity probe\*\*** | RM Young 41382 in aspirated enclosure | Ambient relative humidity at 2 m | 8/2013-6/2017 |
| **Barometer** | Setra 270 | Ambient pressure | 9/2007-6/2017 |
| **Chemical Filters*** | NOAA/ PMEL | Common aerosol ions: $SO_4^{-2}$, $NH_4^+$, $Cl^-$, $Br^-$, $Na^+$, $K^+$, $Ca^+$, $Mg^+$, $NO_3^-$, MSA, and oxalate | 4/2002-05/2006 |

* These measurements measured at two size cuts: $PM_1$ and $PM_{10}$.
† Correction methods used for these instruments are outlined in Anderson and Ogren (1998) for the nephelometer and Bond et al. (1999) and Ogren (2010) for the PSAP and CLAP
**Ambient RH measurements prior to August 2013 were invalid.

*2.1. Aerosol inlet system*

The aerosol inlet and system at THD are the standard NOAA design and have been described in detail in other papers (Sheridan et al., 2001; Delene and Ogren, 2002; Sherman et al., 2015) so only a brief description is given here.  The sampling stack was 10 m tall, constructed of 8" i.d. PVC pipe. A rainhat prevented precipitation from entering sample stack – the rainhat design was changed in March 2010.  A total airflow of ~1000 lpm entered the stack. Most of this airflow (850 lpm) was excess air while the remaining 150 lpm, the sample flow, was pulled through a gently heated 2" stainless steel pipe at the base of the sampling stack and fed into a 5-port flow splitter. The heating applied to the stainless-steel tube was used to lower the relative humidity (RH) of the sample to comply with Global Atmosphere Watch (GAW) sampling protocols of keeping sample RH below 40% (WMO, 2016).  At THD, 30 lpm of the splitter sample flow was routed to the aerosol optical properties measurement system, 30 lpm went to the chemical filter measurements, and 30 lpm was used to monitor the RH in the stainless-steel inlet and control the heat to lower that RH when necessary.  The remaining two lines (three lines after discontinuation of the



chemical measurements in 2006) of 30 lpm flow were not used and went directly from the flow splitter to the pump

box.

For the aerosol optical property measurements, the 30 lpm of the sample flow was further warmed by a

secondary heater when necessary, to maintain the system relative humidity at less than 40%. The average

temperature downstream of the secondary heater was $22 \pm 4$ °C during site operation ($20 \pm 3$ °C, $20 \pm 3$ °C, $23 \pm 5$

°C, and $22 \pm 3$ °C in the winter, spring, summer, and fall respectively). Downstream of the secondary heater, a

switched impactor system provided size-segregated measurements. Sample air first flowed through a Berner-type

multijet cascade 10 μm impactor (Berner et al., 1979) (~7 μm aerodynamic diameter). An electronic ball valve was

used to switch between a direct sample line to the instruments and a second Berner-type impactor with a size cut of

1 μm (~0.7 μm aerodynamic diameter). Here, the 10 μm size cut is referred to as the total size cut or $PM_{10}$ while the

1 μm size cut is referred to as the sub-micron size cut or $PM_1$. The switching between $PM_{10}$ and $PM_1$ size cuts

occurred every 6 min between April 2002 and May 2004, every 30 min between May 2004 and March 2006 to

optimize humidograph operations, and then back to switching every 6 min after the humidograph was removed until

station closing in June 2017.

*2.2. Aerosol chemical composition*

For the first four years of measurements at the site an additional 30 lpm of the flow from the flow splitter was

pulled through a filter carousel system used to collect aerosol on filters for chemical analysis. The filter carousel

system used the same $PM_{10}$ and $PM_1$impactors as the aerosol optical system. Results from an identical system were

described in Quinn et al. (2002). The filters were analyzed by Pacific Marine Environmental Labs (PMEL). More

details of their analysis can be found in Quinn et al. (2000). The samples were analyzed using ion chromatography

for major ionic species including: $SO_4^{2-}$, $NH_4^+$, $Cl^-$, $Br^-$, $Na^+$, $K^+$, $Ca^+$, $Mg^+$, $NO_3^-$, methanesulfonic acid ($MSA^-$) and

oxalate. Aerosol mass concentrations for both $PM_1$ (daily) and for PM between1 μm < Dp < 10 μm (weekly) were

obtained. The $PM_1$ data were then averaged over the weekly sampling time of the 1 – 10 μm PM samples, and the

two masses were added together to achieve a $PM_{10}$ mass that could be compared to the optical data. It is important to

note that some fraction of the $NO_3^-$ mass may have been driven off through the heating of the sampled air described

above, and therefore could be underestimated in this work (Bergin et al., 1997). For the first ~1.5 years of the



program, local contamination episodes were identified when the wind was from the east and southeast sector (specifically 56-186º) and the chemical filtering system was bypassed. In addition to wind direction particle concentration from CN was used to identify periods of local contamination (N > 8000 cm$^{-3}$) (Sheridan et al., 2016).

Chemical sampling was switched off during contamination periods. Identification of contamination using wind sector was stopped on Oct 7, 2003. It was determined that the air flow around the site was heavily influenced by the local topography and was leading to elimination of non-contaminated data. Ion concentrations were compared to elemental measurements made with a rotating drum impactor that operated at THD during the spring of 2002 (**Section S1**). Additionally, simple outlier testing was done on the filter data within monthly and seasonal analysis

based on the interquartile range (IQR). Data points were defined as outliers and were not included in analysis if they fell outside the range of Q25 – (IQR*1.5) to Q75 + (IQR*1.5), where Q25 and Q75 represent the 25$^{th}$ and 75$^{th}$ percentiles.

Contributions of non-sea salt (nss) ion concentrations were calculated following the relationships outlined by Virkkula et al. (2006):

$$\text{Sea Salt} = \text{Cl}^- + 1.47\,(\text{Na}^+) \tag{1}$$

$$nss\,\text{SO}_4 = \text{SO}_4^{2-} - 0.25\,(\text{Na}^+) \tag{2}$$

$$nss\,\text{K} = \text{K}^+ - 0.038\,(\text{Na}^+) \tag{3}$$

$$nss\,\text{Mg} = \text{Mg}^{2+} - 0.12\,(\text{Na}^+) \tag{4}$$

$$nss\,\text{Ca} = \text{Ca}^{2+} - 0.036\,(\text{Na}^+) \tag{5}$$

These relationships are based on the typical major ion concentrations in seawater. As these are calculated values, there are the possibilities for negative values in ion concentrations, instances of this have been highlighted below.

*2.3. Aerosol number concentration*

A butanol-based condensation nuclei (CN) counter (Model 3760, TSI Inc., Minneapolis, MN) was used to

measure the number concentration of particles with diameter > 0.14 µm. Between April 2002 and November 2005, the CN inlet was a separate ¼" stainless steel sample line that ran from the top of the 10 m stack directly to the instrument. The flow in this line was ~10 lpm, where 1 lpm was the CN sample flow and the remaining 9 lpm were used for the counterflow Nafion drier to remove water vapor from the sample line. This was critical as water vapor



is soluble in butanol and as the butanol picks up water the measurement quality degrades. In 2005, the CN sample

line was changed to pick off flow directly from the optical measurements line upstream of the secondary heater and

impactor box.

*2.4. Absorption and scattering measurements*

Optical measurements consisted of aerosol absorption coefficients, measured by two different Particle Soot

Absorption Photometers (PSAP; Radiance Research, 1-wavelength or 3-wavelength) and a Continuous Light

Absorption Photometer (CLAP; NOAA; Ogren et al., 2017), and aerosol total and back scattering coefficients,

measured by an integrating nephelometer (Model 3563, TSI Inc.) (**Table 1**).   The aerosol absorption instruments

were active during different times over the course of the 15-year measurement period.  No overlap measurements

exist for the 1- and 3-wavelength PSAPs in the field, however, comparison between the 3-wavelength PSAP and

CLAP absorption measurements during their overlap period at THD showed a high level of agreement (Slope = 1.06

and $R^2$ = 0.98; Ogren et al., 2017). The CLAP sampled off the nephelometer blower block as described in Ogren et

al. (2017) while both PSAPs sampled off a pickoff ahead of the nephelometer inlet.  All three absorption instruments

were operated at a flow rate of 1 slpm.  Filter spot sizes for the three instruments are comparable and so this

consistent flow ensured similar face velocities and minimized potential discrepancies due to differences in particle

penetration depths (Müller et al., 2011).  All absorption measurements were corrected using the scheme reported by

Bond et al. (1999) for sample area, flowrate, and non-idealities in the manufacturer's calibration.  Additionally, for

the 3-wavelength absorption instruments, the method outlined by Ogren (2010) was used to correct the spectral

absorption measurements. RH within the PSAP and CLAP instruments was not measured, however the PSAP had an

aftermarket heater (installed October 2009) and the CLAP was maintained at 39 C to minimize RH effects on filter

(Ogren et al., 2017).

The bulk of the 30 lpm sample flow exiting the switched impactors was sampled by an integrating

nephelometer. The TSI nephelometer operates at wavelengths of 450, 550, and 700 nm, over an angular range of 7-

degrees (total scatter) and 90-170 degrees (backscatter).  The nephelometer scattering measurements were

corrected for angular truncation and other instrument non-idealities using the method described by Anderson and

Ogren (1998). The $PM_1$ and $PM_{10}$ nephelometer data were corrected using the Anderson and Ogren (1998) sub-




micron and total corrections, respectively (see their Table 4). An insulating jacket was installed on the nephelometer in December 2012 to help lower the RH inside the instrument which was monitored through an internal sensor.

*2.5. Calculated parameters from optical measurements*

Aerosol optical parameters were calculated from the measurements of spectral aerosol scattering and absorption coefficients (**Table 2**). These parameters are often used to provide insight into the characteristics of the aerosol such as size and composition. For example, Collaud Coen et al. (2007) described how the backscattering fraction (BFR) and scattering Ångström exponent (SAE) are sensitive to different parts of the aerosol size distribution while Schmeisser et al. (2017) showed how the relationship between SAE and absorption Ångström exponent (AAE) can

act as a proxy for the composition of the aerosol. Both SAE and AAE were calculated using wavelength pairs rather than fitting across all three wavelengths. For SAE the blue and green wavelengths (i.e., 450 and 550 nm) were used as there have been some reported issues for the red scattering measured by TSI nephelometers (Collaud Coen et al., 2020). For AAE the blue and red wavelengths (i.e., 450 and 700 nm) were used. These wavelengths were adjusted from the native wavelengths of the PSAPs/CLAP to match the wavelengths used by the nephelometer (**Table 1**).

In addition, an effective black carbon mass concentration ($M_{eBC}$) was calculated from the PSAP's measured absorption coefficient ($\sigma_{ap}$). This is referred to as an effective mass concentration because its calculation requires the assumptions that (1) all of the measured absorption is from black carbon and (2) all black carbon particles have uniform light absorption efficiencies. This calculation also requires assuming a value for the mass absorption cross section (MAC); frequently 10 m$^2$ g$^{-1}$ is used. Bond and Bergstrom (2006) suggest that that a MAC of 10 m$^2$ g$^{-1}$ is too

high for freshly emitted light absorbing carbon, and through synthesis of available measurements they suggest an average MAC value of 7.5 ± 1.2 m$^2$ g$^{-1}$ for these aerosols. However, Bond and Bergstrom (2006) acknowledge that ambient and aged aerosol may have larger MAC values. For this work the MAC of 10 m$^2$ g$^{-1}$ was used, recognizing that this could be an overestimate.




**Table 2**: Derived aerosol optical properties using measured scattering and absorption coefficients

| Parameter (Symbol) | Equation | Eq. # | Instrument(s) Used |
|---|---|---|---|
| Single scattering albedo (SSA)* | $SSA = \dfrac{\sigma_{sp}}{(\sigma_{sp} + \sigma_{ap})}$ | 6 | Nephelometer and PSAP/CLAP |
| Scattering Angstrom exponent (SAE)* | $SAE = \dfrac{\log\left(\frac{\sigma_{sp,\lambda 1}}{\sigma_{sp,\lambda 2}}\right)}{\log\left(\frac{\lambda_1}{\lambda_2}\right)}$ | 7 | Nephelometer |
| Absorption Angstrom exponent (AAE)* | $AAE = \dfrac{\log\left(\frac{\sigma_{ap,\lambda 1}}{\sigma_{ap,\lambda 2}}\right)}{\log\left(\frac{\lambda_1}{\lambda_2}\right)}$ | 8 | PSAP/CLAP |
| Backscattering Fraction (BFR)* | $BFR = \dfrac{\sigma_{bsp}}{\sigma_{sp}}$ | 9 | Nephelometer |
| Asymmetry parameter (g) | $g = -7.143889\,(BFR^3) + 7.464439\,(BFR^2) - 3.96356\,(BFR) + 0.9893$ | 10 | Nephelometer |
| Sub-micron scattering fraction (Rsp)* | $Rsp = \dfrac{\sigma_{sp,1\mu m}}{\sigma_{sp,10\mu m}}$ | 11 | Nephelometer |
| Sub-micron absorption fraction (Rap)* | $Rap = \dfrac{\sigma_{ap,1\mu m}}{\sigma_{ap,10\mu m}}$ | 12 | PSAP/CLAP |
| Scattering enhancement factor (*f*(RH))* | $f(\mathrm{RH}) = \dfrac{\sigma_{sp}\ (RH = 85\%)}{\sigma_{sp}\ (RH = 20 - 40\%)}$ | 13 | Nephelometer with humidity conditioning system |
| Effective black carbon mass (eBC) | $M_{eBC} = \dfrac{\sigma_{ap}}{MAC}$ | 14 | PSAP/CLAP |

*In this work SSA, BFR, Rsp, Rap, and *f*(RH) parameters were derived for the 550 nm wavelength. The SAE was derived using the 450 & 550 nm wavelength data and the AAE was derived using the 450 & 700 nm wavelength data.


*2.6. Positive matrix factorization analysis*

To identify source groupings and contributions at THD, positive matrix factorization (PMF) was used through

consideration of the measured ion component masses and the calculated $M_{eBC}$. This study used PMF 5.0 provided by

the United States Environmental Protection Agency (Norris et al., 2014). PMF is a multivariate factor analysis

method based on a weighted least-squares fit, first developed by Paatero and Tapper (1993). The method is a

receptor model which solves the chemical mass balance between measured species and source profiles, as shown in

the equation (Norris et al., 2014):

$$x_{ij} = \sum_{k=1}^{p} g_{ik}\, f_{kj} + e_{ij} \tag{14}$$

where $x_{ij}$ is the concentration of available species (*i* is the sample number and *j* is the number of species), *p* is the

number of factors believed to contribute to the concentrations, $g_{ik}$ is the relative contribution of the *k*th factor to the

*i*th sample, $f_{kj}$ is the concentration of the *j*th species in the *k*th factor, and $e_{ij}$ is the residual value for the *j*th species.



PMF operates under the constraint that none of the samples can have significantly negative source

contributions and derives factor contributions and profiles by minimizing the objective function $Q$ (Paatero, 1997;

Paatero and Tapper, 1994):

$$Q = \sum_{i=l}^{n} \sum_{j=1}^{m} \left[ \frac{x_{ij} - \sum_{k=1}^{p} g_{ik} f_{kj}}{u_{ij}} \right]^2 \tag{15}$$

Here $u_{ij}$ is the associated uncertainty for the sample $x_{ij}$. The associated error for ion masses were calculated based on

the uncertainty analysis provided by Quinn et al. (2000). Any species that were marked as weak in the analysis had

an additional 3% error applied.

Three error estimation methods were used to validate the PMF solutions: displacement analysis (DISP),

bootstrap method (BS), and a combination of the two (BS-DISP). PMF solutions were only accepted and reported if

(1) the observed drop in $Q$ for DISP was below 0.1% and no factor swaps occurred for $dQ_{max} = 4$, (2) from 100

bootstrap runs all factors had a mapping of $\geq$ 90%, and (3) if the observed drop in $Q$ for BS-DISP was below 0.5%

(Brown et al., 2015; Paatero et al., 2014).

*2.7. Humidograph measurements*

From April 2002 – March 2006, a scanning humidity conditioning system (Sheridan et al., 2001) and second

nephelometer were operated in series with the first nephelometer in order to determine the scattering enhancement

factor $f$(RH) as a function of relative humidity (RH). The aerosol exited the first nephelometer which was operating

at low RH (so-called 'dry') conditions into a humidity conditioner and controller that stepped the sample air stream

through a humidity scan (humidities optimally ranged from ~40% to ~90%) over the course of an hour. The

humidified airstream then entered the second "wet" nephelometer and measurements of total scattering ($\sigma_{sp,wet}$) and

backscattering coefficients ($\sigma_{bsp,wet}$) as a function of relative humidity were obtained. The humidigraph generated

hourly scans of increasing RH with alternating size cuts every 6 min between April 2002 and May 2004. From May

2004 to March 2006 the humidigraph performed 30 min increasing RH scans on the $PM_{10}$ size cut and then 30 min

decreasing RH scan on the $PM_1$ size cut as this allowed better control at the high and low RH conditions. The

scattering enhancement parameter at RH=85% and 550 nm was derived from fitting an equation of the form:



$$f(\text{RH}, \lambda) = \frac{\sigma_{sp}(RH, \lambda)}{\sigma_{sp}(RH_{dry}, \lambda)} \tag{16}$$

We used the THD *f*(RH) dataset (Burgos et al., 2019b) and further details about the data processing to calculate

*f*(RH) are described in the original publication (Burgos et al., 2019a). The value of *f*(RH) presented here is the fitted

value for wet scattering at 85% and 'dry' scattering measured at the RH of the dry nephelometer (**Table 2**). The dry

nephelometer RH ranged from ~20% in the winter to ~40% in the summer, with an annual median dry nephelometer

RH value of 29%.


*2.8. Meteorology*

From 2002-2007 the only meteorological parameters measured were wind direction (WD) and wind speed (WS)

at 10 m. In 2007 a suite of meteorological measurements was added to the site, including ambient temperature at 2

and 10 m, relative humidity at 2 m, and pressure at 2 m, as well as an additional anemometer for wind direction and

speed at 10 m. A 4-year period of overlapping 10 m WD and WS measurements indicated good agreement between

the original and new anemometer and the original anemometer was removed in 2011. Unfortunately, the RH

measurements initiated in 2007 were determined to be invalid between 2007-2012 due to problematic wiring of the

sensor and valid ambient RH data is only available for 2013-2017.

Typical wind directions were the same in summer and winter, but summer tended to be more influenced by flow

from the ocean (the northwest). In contrast, in the winter the wind was more likely to come from the land to the

south and east of the site. The diurnal flow regimes indicated in **Figure S3** are consistent with typical

onshore/offshore flow observed at many coastal sites. The southeasterly airflow pattern is a land breeze (flowing

from land to ocean) - it occurs primarily at night time (8pm-8am) and is directionally consistent with the air coming

from nearby coastal communities and the local harbors. In contrast, the northwesterly flow occurs during the day

with the wind coming primarily from the Pacific Ocean. The wind rarely comes from the northeast sector or the

southwest sector. It should be noted that due to the complex topography near the site, the local wind direction likely

represents very local flow patterns. Monthly and diurnal statistical plots of wind data are included in the

supplemental materials (**Section S2**).





## 3. Results and Discussion

*3.1. Aerosol Chemical Components*

3.1.1. Seasonal Cycles of Ionic Components

Seasonal median $PM_{10}$ concentrations for 2002 to 2006 suggest that the ions most likely to be associated with sea salt (i.e., $Na^+$, $Cl^-$, $Mg^{2+}$, $Ca^{2+}$, $K^+$ and $Br^-$) are lower in the July-September time period and higher November through April (**Fig. 1**). This may be due in part to the lower wind speeds in the summer generating less sea spray. $PM_1$ ions whose primary source is likely to be sea salt ($Na^+$, $Cl^-$, $Mg^{2+}$ and $Ca^{2+}$) exhibit broadly similar seasonal cycles to those observed for the $PM_{10}$ measurements, but there are differences in $PM_1$ and $PM_{10}$ seasonality for other ions (**Fig. 1**). $PM_{10}$ ions associated with marine biogenic activity ($MSA^-$ and oxalate) peak in the summer. The $PM_1$ seasonal cycles for $MSA^-$, $NH_4^+$, and $SO_4^{2-}$ are similar and peak in May-September, presumably due to marine biogenic activity. However, $PM_1$ oxalate exhibits relatively stable concentrations throughout the year. This indicates that $PM_1$ oxalate may come from multiple sources at different times, including biogenic, combustion, and anthropogenic sources (Rinaldi et al., 2011; Saarnio et al., 2010; Andreae, 1983; Baudet et al., 1990; Gao et al., 2003). $PM_1$ total mass, $NO_3^-$, and $K^+$ concentrations peak in the cooler months (October-January) suggesting that these are likely related to anthropogenic activity (i.e., wintertime residential heating). Comparisons between the $PM_{10}$ and $PM_1$ ion concentrations show that $MSA^-$, $NH_4^+$, $SO_4^{2-}$, and oxalate occur primarily in the $PM_1$ size range – their concentrations are often extremely close to the median $PM_{10}$ ion concentrations. Other measured ions, such as, $Na^+$, $Cl^-$, $Mg^{2+}$, and $NO_3^-$ are primarily in the supermicron size range – the $PM_{10}$ concentrations of these ions are at least an order of magnitude higher than their $PM_1$ concentrations – suggesting a link to sea salt.

For both size fractions the undetermined mass (which could be organics, insoluble ions, etc.) makes up between 60-80% of the total mass throughout the year (**Table S1, S2**). The estimated contribution of black carbon is discussed later (**Section 3.1.3**). There is no obvious seasonal cycle in the undetermined mass fraction, however, there is seasonality in the fractional contribution of different ions to total ion mass (**Fig. 2**). Sea salt ions ($Na^+$, $Cl^-$, $Mg^{2+}$, and $Ca^{2+}$) make up the majority of $PM_{10}$ ion mass in all four seasons, however, they only dominate the $PM_1$ ion mass in the winter. In the spring, summer, and fall $SO_4^{2-}$ makes up the largest fraction of $PM_1$ ion mass. Both the $PM_1$ and $PM_{10}$ ion mass fractions of $MSA^-$ are highest in the summer when marine biogenic activity increases. $PM_1$ fractions of $K^+$, $NO_3^-$, and oxalate are highest in the winter and lowest in the summer, however, $PM_{10}$ fraction of $K^+$



is relatively consistent across all seasons and $PM_{10}$ fractions of $NO_3^-$ and oxalate are highest in the summer and lowest in the winter.

Since THD is a marine site, the aerosol chemistry will exhibit a strong sea salt component. It is therefore useful to elucidate the contributions of non-sea salt (nss) ion concentrations relative to those derived from marine sources. The relationships (Eq. 1-5) from Virkkula et al. (2006) were used to calculate concentrations of sea salt and non-sea salt (nss) ion concentrations using $Na^+$ as the reference species (**Section 2.2**). The estimated seasonal cycle of total concentrations for sea salt as well as contributions of nss $SO_4^{2-}$, $K^+$, $Mg^{2+}$ and $Ca^{2+}$ to the total measured concentrations for both $PM_1$ and $PM_{10}$ are shown in **Figure 3**. Sea salt (due to soluble ions) makes up approximately 30% of the identified $PM_{10}$ mass concentration throughout the year but only 5-14% of the $PM_1$ mass concentration. The lowest sea salt contributions to $PM_1$ occur from May to September, with higher sea salt contributions the rest of the year. Virtually all of the $PM_{10}$ $K^+$, $Mg^{2+}$ and $Ca^{2+}$ is attributable to sea salt (by default, $Na^+$ and $Cl^-$ are assumed to be solely due to sea salt). Interestingly, $PM_{10}$ $SO_4^{2-}$ was largely attributable to sea salt in colder months (November – April), however, from May to October the nss $SO_4^{2-}$ fraction dominated the $PM_{10}$ $SO_4^{2-}$ total concentration. This pattern is consistent with increased biogenic activity in the summer as a source of $SO_4^{2-}$. Similar to $PM_{10}$, $PM_1$ $Mg^{2+}$ can be almost entirely attributed to sea salt. In contrast, $PM_1$, $SO_4^{2-}$ and $K^+$ appear to be primarily nss in origin., while $PM_1$ $Ca^{2+}$ is marginally attributable to both sea salt and nss sources.





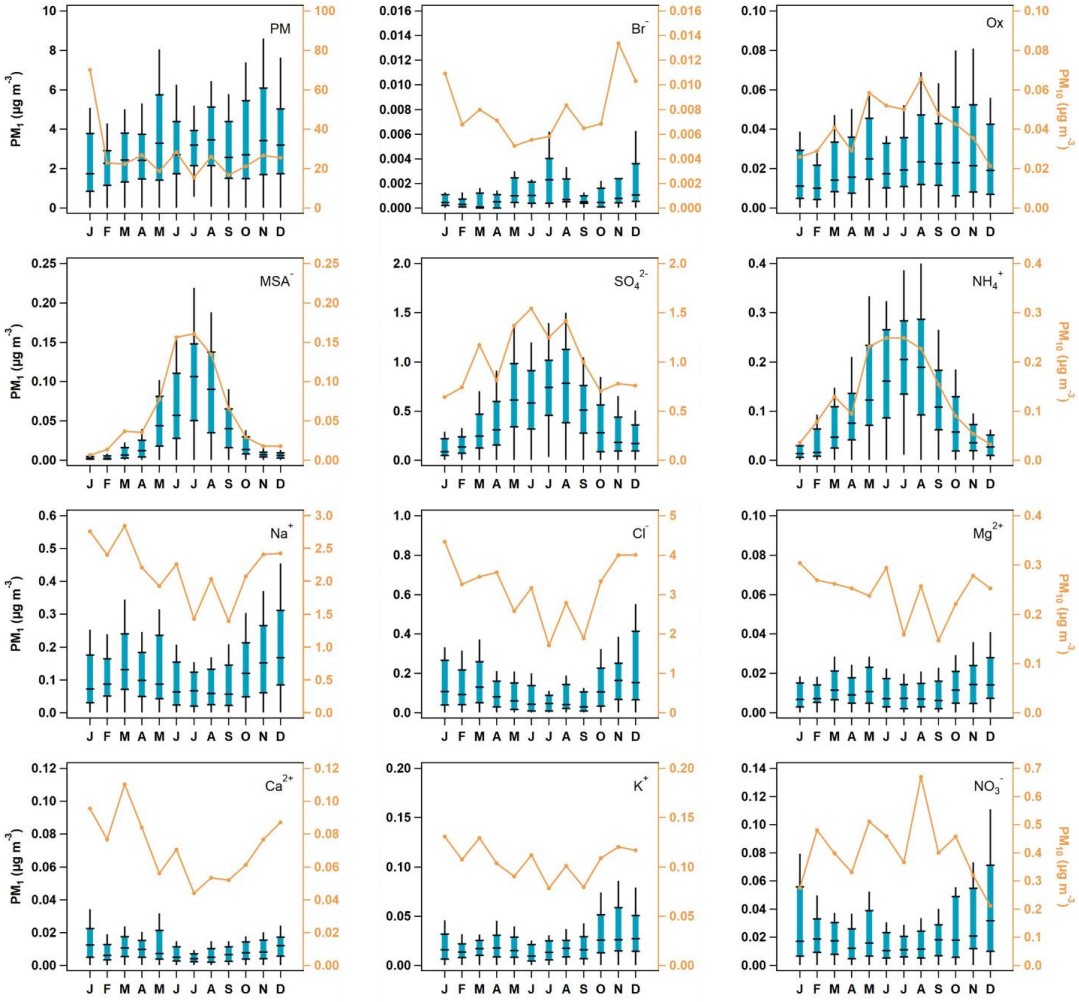

**Figure 1**: Seasonal aerosol mass and ion concentrations; blue box-whiskers represent PM$_1$ aerosol percentiles (5th, 25th, median, 75th and 95th) and the yellow line represents median concentration of PM$_{10}$ particles. The PM$_1$ data scale is on the left axis, while the PM$_{10}$ is on the right which has been color matched to the PM$_{10}$ trace.




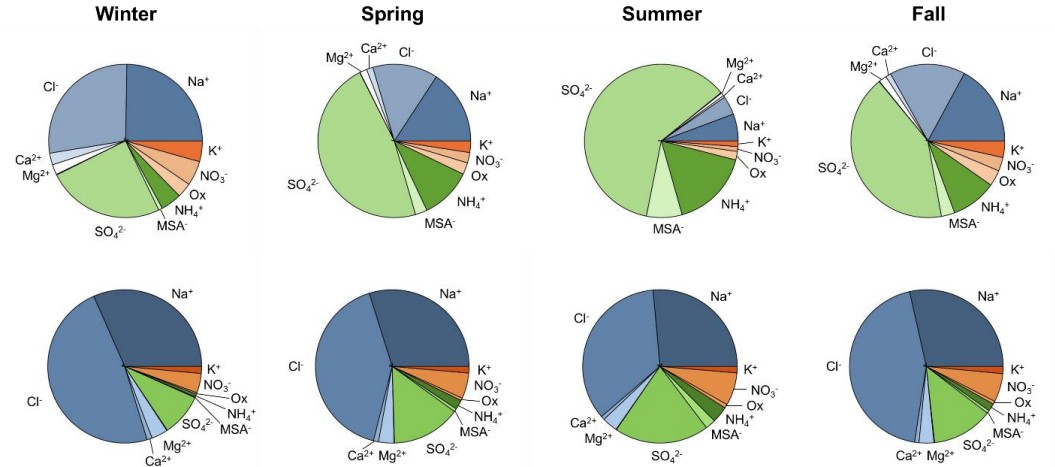


**Figure 2**: Ion mass fractions for PM$_1$ (top row) and PM$_{10}$ (bottom row) in all four seasons. This data is based on total ion mass, not total PM$_1$ or PM$_{10}$ mass. Undetermined mass is not included.





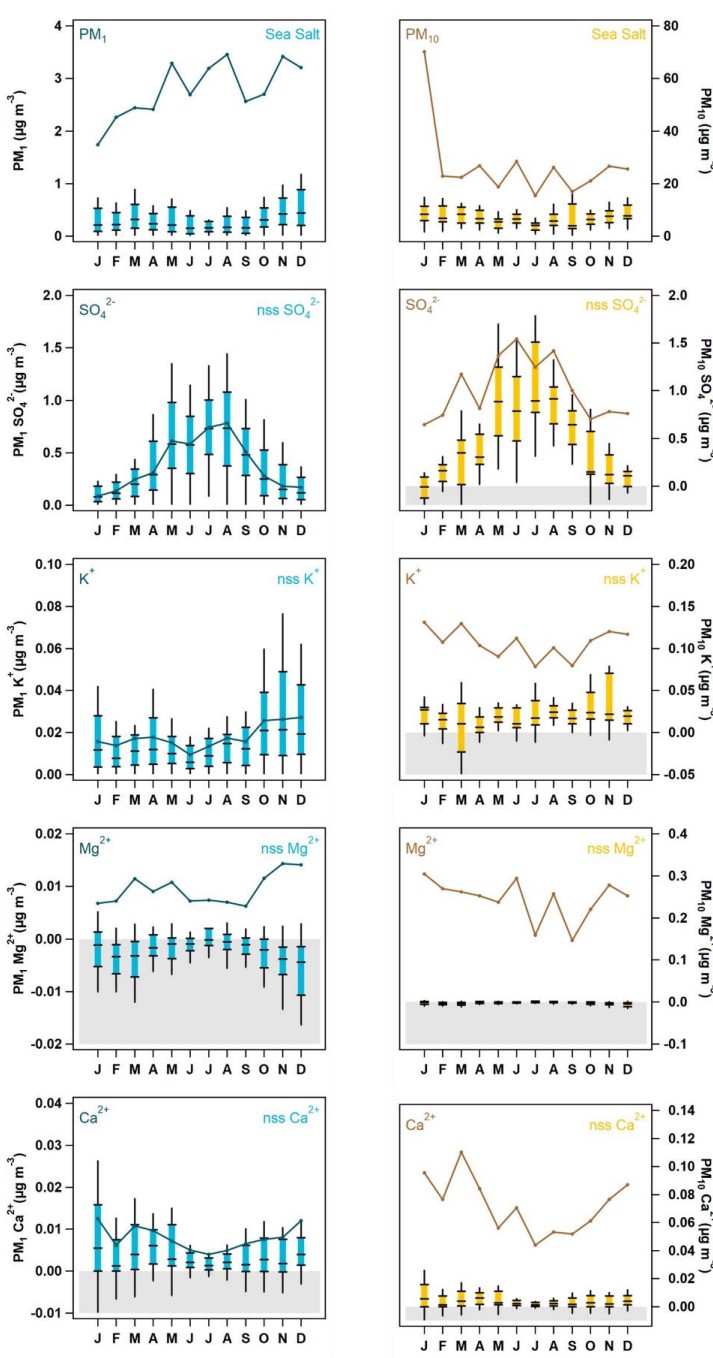

**Figure 3**: Seasonal cycle of PM$_1$ (blue) and PM$_{10}$ (yellow) sea salt and ions with non-sea salt (nss) contributions. The box-whiskers represent the sea salt and nss ion statistics (5$^{th}$, 25$^{th}$, median, 75$^{th}$, and 95$^{th}$ percentiles) while the

darker solid line shows the median total measured mass of the ion relative to the sea salt or nss component over the four-year period. Concentrations that were calculated to be below zero are highlighted in grey on all plots.

3.1.2.  Seasonal ionic relationships as source indicators

We further explored the seasonal relationships between all measured ions using simple linear regression analysis. Following the ion mass fractions, in the winter $PM_1$ total mass has strong correlations ($r > 0.8$) with ions likely to be from sea salt ($Na^+$, $Cl^-$, $Mg^{2+}$, $K^+$, and $Ca^{2+}$) whereas in the summer $PM_1$ has the strongest correlations with ions associated with biogenic activity ($SO_4^{2-}$, $MSA^-$, and oxalate). Sodium ($Na^+$), magnesium ($Mg^{2+}$), and

chloride ($Cl^-$) are always highly correlated ($r > 0.8$), and all three ions have significant relationships ($r > 0.5$) with calcium ($Ca^{2+}$) in all four seasons. Both $Na^+$ and $Mg^{2+}$ correlate with potassium ($K^+$) in the winter, spring, and fall, however $Cl^-$ only has significant relationships with $K^+$ in the winter and fall. Sulfate ($SO_4^{2-}$) is always significantly correlated with methanesulfonic acid ($MSA^-$) and oxalate. However, $MSA^-$ and oxalate only have strong linear relationships in the winter and summer. Additionally, $SO_4^{2-}$ is highly correlated ($r > 0.9$) with ammonium ($NH_4^+$) in

every season, indicating that particles in the area are largely composed of acidic compounds which is consistent with results presented by Allan et al. (2004) who observed a large particulate sulfate to ammonium ratio in the spring of 2002. Ions that are also associated with anthropogenic emissions ($K^+$, nitrate ($NO_3^-$), and oxalate) are consistently correlated in the winter and fall. $K^+$ and oxalate also have a significant relationship in the spring and the two ions both correlate with $NO_3^-$ in the summer. The correlation coefficients for all $PM_1$ ion relationships are listed in the

supplemental (**Table S3**).

The $PM_{10}$ data exhibits fewer significant relationships between ions, which could be the result of a reduced number of available data points due to weekly sampling. However, there are still clear relationships present. Just as with the $PM_1$ data, $Na^+$, $Mg^{2+}$, and $Cl^-$ were always highly correlated ($r > 0.8$), and all three ions had significant relationships ($r > 0.5$) with $Ca^{2+}$ in the spring, summer, and fall. Relationships between $Na^+$, $Mg^{2+}$, $Cl^-$, and $K^+$ were

the most significant ($r > 0.8$) in the winter and fall. $PM_{10}$ $SO_4^{2-}$ is highly correlated ($r > 0.8$) with sea salt ions ($Na^+$, $Cl^-$, $Mg^{2+}$, and $K^+$) in the winter, and in the other months is more consistently correlated with $NH_4^+$, $NO_3^-$, $MSA^-$, and oxalate. This is consistent with our non-sea salt fraction analysis of $SO_4^{2-}$ which showed higher fractions of sea salt $SO_4^{2-}$ in the winter months and lower fractions in other seasons. The correlation coefficients for all $PM_{10}$ ion relationships are listed in the supplemental (**Table S4**).



From these relationships there are clear seasonal source groupings of sea salt ions, biogenic ions, and

anthropogenic ions. These patterns are further supported by changes in ion concentration ratios across seasons. For

instance, the magnitude of the $MSA^-$ to nss $SO_4^{2-}$ mass concentration ratio can discern between the presence of clean

marine air with enhanced biological activity, when the ratio is high, and the existence of anthropogenic sulfate,

when the ratio is low (Savoie and Prospero, 1989). The $PM_1$ $MSA^-$ /nss $SO_4^{2-}$ ratio at THD is highest in the summer,

with median values ranging from $0.1 - 0.15$ (equivalent to molar ratios of 8-12%), and lowest in the winter, with

median values of $0.02 - 0.03$ that are nearly an order of magnitude lower than the summer ratios (**Fig. S4**). This

generally agrees with the average $PM_1$ $MSA^-$ /nss $SO_4^{2-}$ ratio of 0.17 found by Millet et al. (2004) in the springtime

period of the NOAA ITCT 2K2 study. Similarly, the mass concentration ratio of oxalate to nss $SO_4^{2-}$ can provide

evidence for clean marine air or air masses influenced by biomass burning. Zhou et al. (2017) reported oxalate to nss

$SO_4^{2-}$ ratios of $< 0.04$ for clean airmasses and ratios between $0.1 - 0.3$ for biomass burning influenced air. At THD

the $PM_1$ oxalate /nss $SO_4^{2-}$ ratio is lowest in the summer $(0.03 - 0.04)$ and highest in the winter $(0.1 - 0.15)$. This

pattern is also observable in the $PM_{10}$ ion ratios; however, there is larger variability (**Fig. S4**). These ratio values

give further evidence to increases in biogenic sources at THD during the summer and increased influences from

woodburning home heating in the winter (City of Arcata, 2008).


3.1.3. Effective black carbon mass contribution

An effective black carbon mass concentration ($M_{eBC}$) was calculated for the data using the absorption

coefficient ($\sigma_{ap}$) measured by the PSAP. Following the seasonal cycle in absorption (**Section 3.2**), $M_{eBC}$ was highest

in the colder months (September through February) and lowest in the summer (**Fig. S6a**). $M_{eBC}$ also exhibited a

distinct bimodal diurnal cycle (**Fig. S6b**), indicating regular sources in the mornings and evenings which is likely

from vehicular and/or maritime traffic. These seasonal observations are discussed further below (**Section 3.2**). For

$PM_1$, $M_{eBC}$ accounted for $1 - 2.5\%$ of the total mass, with larger fractions in the winter and fall (**Table S1**). In

contrast, the contribution of $M_{eBC}$ to $PM_{10}$ was negligible $(> 0.5\%)$ in all seasons (**Table S2**).

Linear regression analysis of $M_{eBC}$ against the other ions in both size fractions was done for the seasonal

and monthly data. When grouped by season $PM_1$ $M_{eBC}$ only correlated with oxalate in the winter, however,

segregating the data by month showed more relationships of interest. $M_{eBC}$ correlated with $PM_1$, $SO_4^{2-}$, $K^+$, $NO_3^-$,

and oxalate in December, and surprisingly also had a significant relationship with $MSA^-$ during this time. The



correlations with $SO_4^{2-}$, $K^+$, and oxalate were also present in January, and other significant correlations appeared for

$K^+$ and oxalate in other months (**Table S5**). For both $SO_4^{2-}$ and $K^+$, the correlation with $M_{eBC}$ generally increased

when only the non-sea salt (nss) fraction of the ion concentrations was considered (**Table S5**). Similar patterns were

observed in the $PM_{10}$ data (**Table S6**), with significant correlations between $M_{eBC}$ and $PM_{10}$, $SO_4^{2-}$, $K^+$, $NO_3^-$, $MSA^-$,

and oxalate being most common in the October – February period. Winter and fall correlations with both $SO_4^{2-}$ and

$K^+$ were again much improved when only the nss fractions were considered. These relationships provide further

evidence for the presence of anthropogenic combustion sources in the winter and fall, which aligns with the ion

seasonal cycles and correlations discussed previously.

### 3.1.4. Source identification using PMF analysis

To confirm the ionic source groupings indicated by the linear relationships described above, positive matrix

factorization (PMF) was used (Norris et al., 2014). PMF analysis on the entire $PM_1$ ion data set, without segregation

by season, resulted in the identification of only two factors based on the quality control criteria used (**Section 2.3**).

However, separating the data by seasons resulted in better resolution of source factors.  It should be noted that this

seasonal separation did lead to different weighting of ionic compounds among seasons, which is summarized in the

supplemental along with results of the PMF error analysis for each season (**Section S4**). Factor analysis (using an

alternative method to the PMF analysis presented here) was performed previously by Millet et al. (2004) on hourly

volatile organic compound (VOC) measurements collected at THD during the NOAA ITCT 2K2 study (19 April –

22 May 2002). Within the higher resolution ITCT springtime data five factors were identified: (1) local

anthropogenic emissions predominately from fossil fuel use, (2) oxygenated compounds from a variety of sources,

(3) long-lived anthropogenic emissions, (4) compounds affected by local atmospheric mixing, and (5) local

terrestrial biogenic emissions. Both factors 1 and 4 were strongly influenced by local meteorology, indicating that

405    loss of this resolution could be limiting our factor identification here. While PMF analysis was performed for each

seasonal data set, both the spring (March – May) and fall (September – November) data resulted in two-factor

solutions that could not be validated based on subsequent error analysis and therefore are not included in the

discussion here.

For the winter (December – February), a three-factor solution was achieved (**Fig. 4a**). The first factor is

410    determined to be sea salt, given the high contribution of $Na^+$, $Mg^{2+}$, and $Cl^-$. The second factor appears to be



biogenic with significant contributions from $SO_4^{2-}$, $MSA^-$, and $NH_4^+$ along with a small fraction of oxalate. Finally, the third factor is the anthropogenic combustion source as shown by the high fractions of oxalate, $K^+$, and $M_{eBC}$. These factor profiles are consistent with the significant ion relationships identified in the winter. Additionally, the sea salt, biogenic, and anthropogenic/ combustion factors were estimated to make up approximately $40 \pm 30\%$, $30 \pm 20\%$, and $30 \pm 20\%$ of the ion mass throughout the winter (**Fig. 4a**) which is fairly consistent with our direct ion mass calculations (**Fig. 2**). PMF analysis of the summer ion data (June – August) resulted in a two-factor solution (**Fig. 4b**). Similar to the winter, in the summer the first factor is sea salt, with large contributions from $Na^+$, $Ca^{2+}$, and $Mg^{2+}$, while the second factor is biogenic, dominated by contributions from $SO_4^{2-}$, $MSA^-$, and $NH_4^+$ and a significant contribution from oxalate. Sea salt and biogenic factors made up approximately $40 \pm 30\%$ and $60 \pm 30\%$ of the ion mass in the summer respectively. These factors agree strongly with the summertime linear correlations and generally follow ion mass contributions (**Fig. 2**)

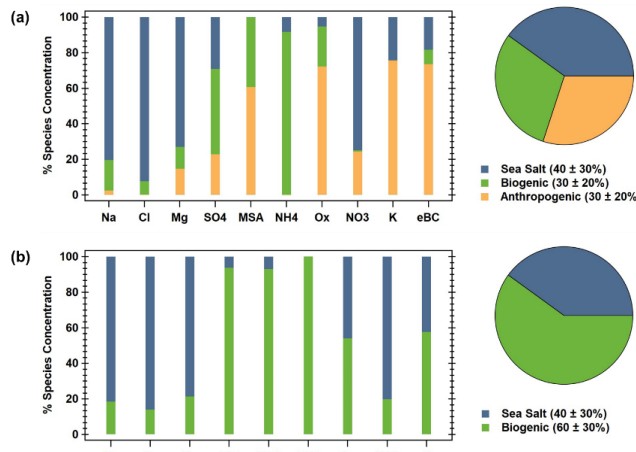

**Figure 4:** PMF factor information for the (**a**) winter and (**b**) summer. The bar charts show the ion species percent contribution to each factor (sea salt is blue, biogenic is green, and anthropogenic is orange), and the pie charts show the percent contribution (listed in the figure legends) of each factor to the seasonal $PM_1$ mass.

*3.2. Aerosol Optical Properties*

3.2.1. Seasonal cycles of optical properties

Seasonal cycles for the optical parameters are consistent across the 15 years of measurements, although the amount of aerosol (as indicated by scattering and absorption) decreases from the start to end of the measurements.



General trends in aerosol optical properties at THD over time are reported on further by Collaud Coen et al. (2020).

There is an obvious seasonality for most of the aerosol particle properties (**Fig. 5**), and the PM$_{10}$ aerosol exhibits

similar seasonal patterns to those observed for the PM$_1$ data. Amount of aerosol – as represented by number

concentrations (CN), light absorption, and light scattering (**Fig. 5a, b, c**) – is highest in fall and winter (September

through January) with October and November having the highest aerosol loading and summer months (June, July,

and August) tending to have the lowest loading. The aerosol is darkest (i.e., single scattering albedo (SSA) is lowest)

in the fall and winter (**Fig. 5e**). A similarly-timed decrease in SSA is noted at other (non-marine) North American

sites, where it is attributed to less production and/or more efficient removal of large, highly scattering particles

during early autumn (Sherman et al., 2015).

The seasonality of aerosol size distribution at THD can be indirectly observed through the monthly cycles

in the scattering Angstrom exponent (SAE), submicron scattering fraction (R$_{sp}$), and the backscatter fraction (BFR).

The SAE (**Fig. 5f**) and sub-micron scattering fraction (R$_{sp}$) (**Fig. 5d**) have similar seasonal patterns. Both SAE and

Rsp are highest in summer, indicating a larger contribution from PM$_1$ particles, and lowest in the winter,

demonstrating an increase in large particles. The median values of PM$_{10}$ SAE (calculated from the 550 & 700 nm

wavelength pair) at THD are always below 1.5 and the median values of Rsp are always below 0.6 indicating a

significant contribution from super-micron aerosol (likely sea salt) throughout the year. BFR (**Fig. 5i**) exhibits the

opposite pattern of SAE and Rsp – it is lowest in summertime suggesting a shift toward larger accumulation mode

particles in the warmer months. BFR and SAE are sensitive to different parts of the aerosol size distribution

(Collaud Coen et al., 2007) so the summertime shift to a higher PM$_1$ particle contribution (increased SAE) in

conjunction with more, larger accumulation mode particles (decreased BFR) could suggest a narrowing of the size

distribution rather than inconsistency in the measurements. In contrast, the lower SAE and higher BFR values in

winter could suggest the size distribution is broader during that time of year. These relationships could also be

attributed to changes in the number or relative importance of different aerosol size modes as multi-modal size

distributions can complicate interpretation of SAE values (Schuster et al., 2006).

Little variation in the submicron absorption fraction (Rap, **Fig. 5g**) is observable throughout the year. Rap

values suggests 80-90% of absorption is PM$_1$ aerosol. In contrast, the absorption Angstrom exponent (AAE,

calculated from the 450 & 700 nm wavelength pair, **Fig. 5h**) exhibits a strong seasonal cycle with the lowest values

(AAE < 1) observed in summer while larger values (AAE ≥ 1.5) are found in winter. AAE values less than one





have been associated with large, non-absorbing particles (Schmeisser et al., 2017), which is consistent with the

increased SSA values in the summer indicating the presence of primarily scattering aerosol (e.g, sea salt). As noted

above, $PM_{10}$ SAE is always less than 1.5 and, in general, values of SAE in the summer are elevated which also

supports the presence of smaller non-absorbing aerosols in the summer.

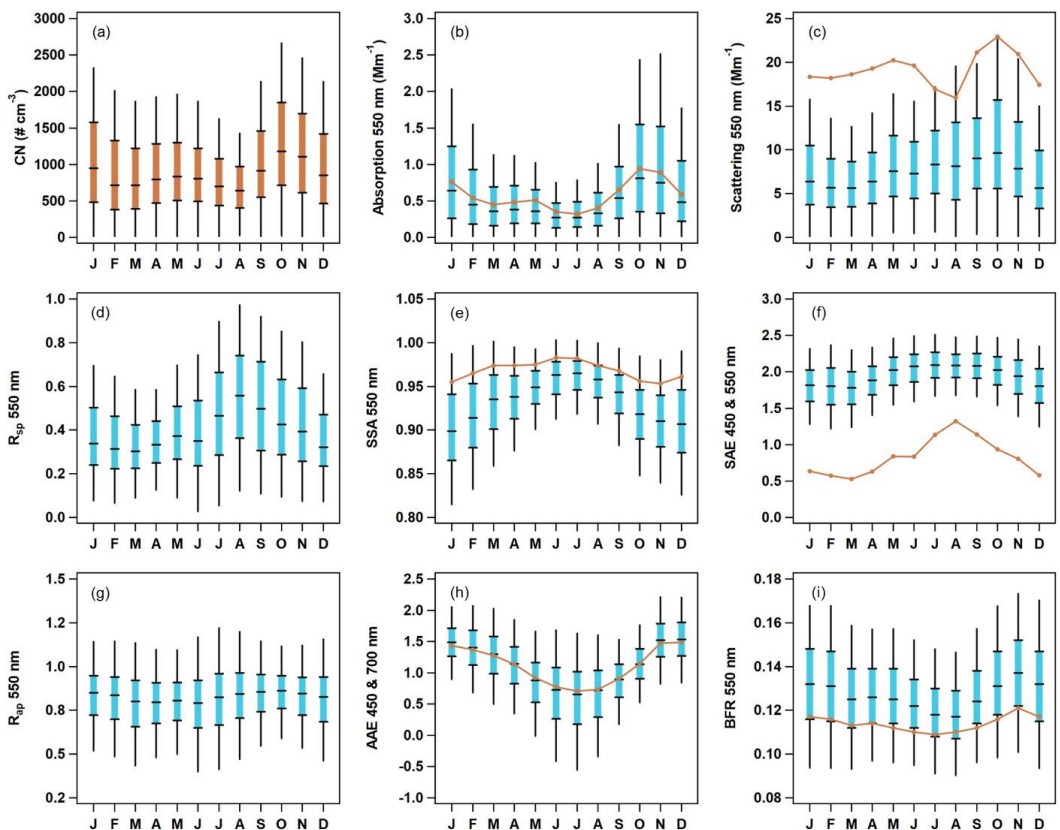

**Figure 5**: Seasonal cycles of aerosol optical properties at THD. The blue box-whiskers represent $PM_1$ percentiles (5th, 25th, median, 75th and 95th) and the orange lines represent median values of $PM_{10}$ optical data. Number concentration data **(a)** is in orange as it includes both $PM_1$ and $PM_{10}$. The optical parameters presented are: **(b)** absorption at 550 nm, **(c)** scattering at 550 nm, **(d)** submicron scattering fraction, **(e)** single scattering albedo for 550 nm, **(f)** scattering Ångström exponent for 450 & 550 nm, **(g)** submicron absorption fraction, **(h)** absorption
Ångström exponent for 450 & 700 nm, and **(i)** backscatter fraction at 550 nm.



### 3.2.2. Diurnal cycles of optical properties

Two different patterns are observable in the diurnal cycles of aerosol optical property data for all 15 years

of sampling at THD (**Fig. 6**). The first pattern is bimodal, with peaks occurring at ~9 am and ~9pm PST. This

pattern is most obvious for variables related to aerosol loading (i.e., CN concentration, absorption, and scattering;

**Fig. 6a, b, c**) but is also seen in the diurnal cycles of the single scattering albedo (SSA; **Fig. 6e**) and absorption

Ångström exponent (AAE; **Fig. 6h**). The diurnal cycle for SSA has minima coinciding with the aerosol loading

maxima. The timing of the diurnal SSA decrease and loading increase suggests a potential

anthropogenic/combustion influence, likely vehicular and/or maritime traffic. This bimodal pattern occurs during all

seasons, although the timing of the peaks shifts slightly as a function of season (not shown). The second pattern

observed in the diurnal cycle plots is a single broad peak in the morning with a maximum near 7am PST. This broad

peak is seen in the plots depicting variables related to particle size, (e.g., SAE, $R_{sp}$, and BFR; **Fig. 6f, d, i**). This

suggests smaller particles are most prominent in the early morning (6-8 am PST) as indicated by increases in SAE,

$R_{sp}$, and BFR at that time.  Just like the bimodal cycles discussed earlier, there is some seasonal variation in the

timing and amplitude of the broad peak (not shown). For most seasons the peak occurs at a similar time in the

morning, but in the summer the amplitude of the broad peak is largest and occurs around noon (possibly related to

marine biogenic activity). The submicron absorption fraction ($R_{ap}$; **Fig. 6g**) has negligible diurnal variability,

suggesting little change in the sources of absorbing aerosol throughout the day.


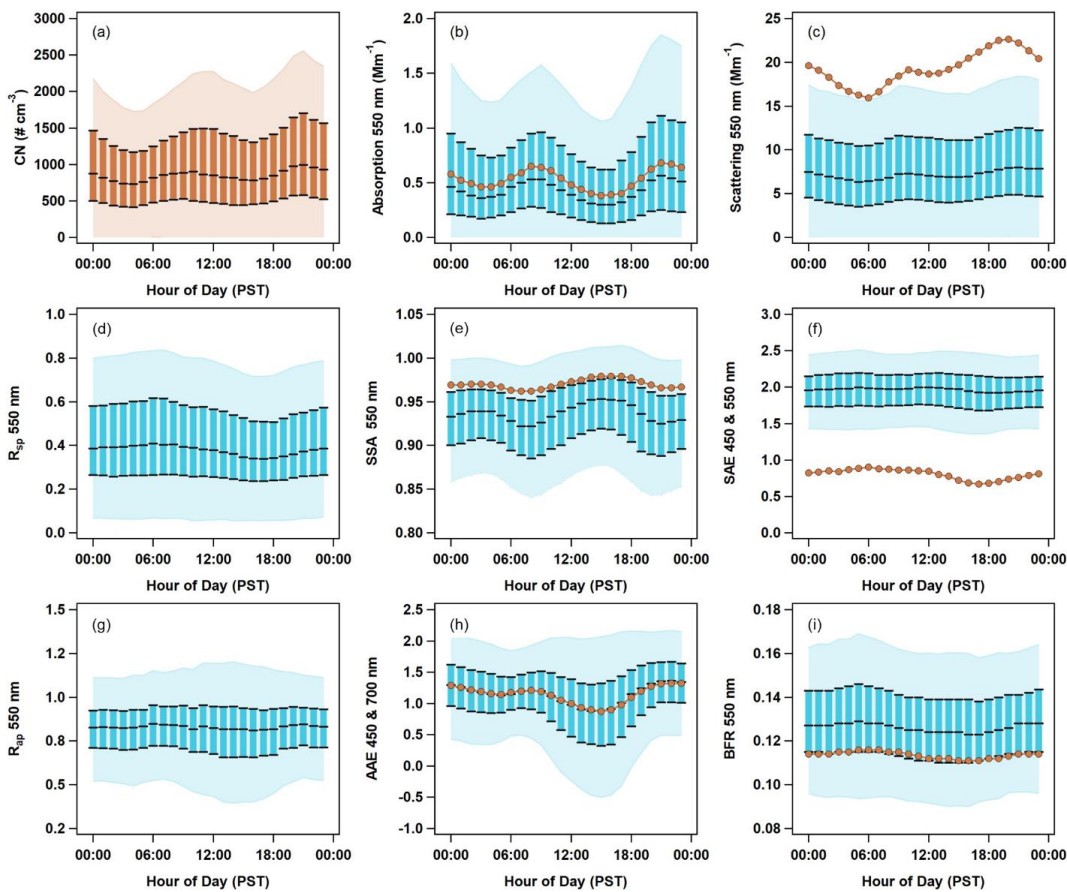

**Figure 6**: Diurnal cycles of aerosol optical properties at THD. Data are plotted against local time (Pacific Standard Time). PM$_1$ optical data is shown in blue with the light blue shading representing the $5^{th}$ to $95^{th}$ percentile and the blue boxes representing the interquartile range ($25^{th}$ to $75^{th}$ percentile) and median. PM$_{10}$ median values are shown as darker orange lines and markers. Number concentration data **(a)** is in orange as it includes both PM$_1$ and PM$_{10}$. The optical parameters presented are: **(b)** absorption at 550 nm, **(c)** scattering at 550 nm, **(d)** submicron scattering fraction, **(e)** single scattering albedo for 550 nm, **(f)** scattering Ångström exponent for 450 & 550 nm, **(g)** submicron absorption fraction, **(h)** absorption Ångström exponent for 450 & 700 nm, and **(i)** backscatter fraction at 550 nm.




### 3.2.3. Temporal cycles in $f$(RH)

Scattering enhancement parameters ($f$(RH) = 85%) are relatively constant throughout the year with a value of ~ 2. In what follows $f$(RH) will refer to the enhancement at RH = 85% unless otherwise stated. The PM$_1$ $f$(RH) values are generally higher than those of the PM$_{10}$ size cut. This is consistent with Zieger et al. (2010) who used Mie calculations to show that, for a given aerosol composition, $f$(RH) will decrease as the mode diameter of the size distribution increases (their Fig. 9). This is due to the fact that the scattering efficiency factor (Q$_{scat}$) is more sensitive to changes in particle size at smaller diameters where the Mie curve is steeper. At larger particle sizes the Mie curve is relatively constant. Values for $f$(RH) observed at THD are also consistent with literature values of marine aerosol hygroscopicity synthesized in Titos et al. (2016).

Surprisingly neither the average seasonal nor the diurnal cycles in the $f$(RH) (**Fig. 7a, b**) reflect the changes in chemistry, particularly the higher levels of SO$_4^{2-}$, NH$_4^+$, and MSA$^-$ observed in the summer (**Fig. 1, 2**) and the local anthropogenic sources suggested by diurnal cycles in the optical properties (**Fig. 6**). Large fractions of sea salt could explain the stability in the PM$_{10}$ $f$(RH). This may also, in part, be due to the seasonal cycle of RH values in the dry nephelometer. Zieger et al. (2014) showed that marine aerosol may pick up significant amounts of water at RH values below 40%. This could result in the seasonal cycle being masked due the 'dry' aerosol scattering not actually being dry at certain times of year. However, the prevalence of certain wind directions in the data set clearly affects the overall seasonal cycles in $f$(RH). Grouping the $f$(RH) by four major wind quadrants – northeast (0 – 90°), southeast (90 – 180°), southwest (180 – 270°), and northwest (270 – 360°) – allows seasonal and diurnal cycles to emerge (**Fig. 7c, 7d, S8, S9**).

Both seasonal and diurnal cycles of PM$_1$ $f$(RH) are weakest in the northwest and southeast quadrants, which dominated measurements at this site (**Fig. S3, S8, S9**). The northeast sector shows patterns that indicate local anthropogenic influences, with $f$(RH) being lowest in the fall and winter (Oct – Feb) and highest in the late spring and early summer (April – June). In the diurnal cycle there is a decrease in $f$(RH) at ~8:00 AM (PST), after which the $f$(RH) steadily increases until ~16:00 PM (PST), which correlates with patterns in the absorption coefficient and other optical properties (**Fig. 6**). This is evidence for an increase in hygroscopicity as fresh emissions from morning traffic age throughout the day, with the drop in $f$(RH) later indicating reintroduction of fresh combustion aerosols from afternoon traffic. These diurnal patterns are weaker but still present in the PM$_{10}$ data (**Fig. S9, S10**). Further

evidence of anthropogenic sources in the northeast are discussed later (**Section 3.2.4**). The southwest has similar

seasonal increase of $f$(RH) in the late spring and early summer (April – June), however, this direction does not

exhibit the same decrease in aerosol hygroscopicity in the fall and wintertime. In the diurnal cycle, $f$(RH) is

marginally higher before ~7:00 AM (PST) and after ~17:00 PM (PST).  This decrease in hygroscopicity could

indicate continental influence during the day for winds from this sector, however, this is unlikely as these decreases

do not correlate with decreased SSA or increases in the scattering or absorption Ångström exponent values.

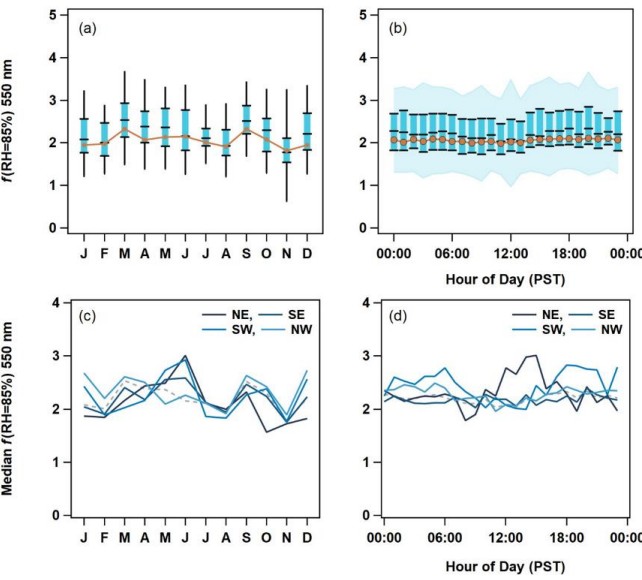

**Figure 7**: **(a)** Seasonal and **(b)** diurnal cycles of $f$(RH) for 550 nm. The $PM_1$ 5th to 95th percentiles are indicated by black lines in (a) and blue shading in (b). The $PM_1$ 25th to 75th interval is highlighted by blue boxes and black markers. Finally, $PM_1$ median values are black markers and $PM_{10}$ median values are the orange traces. The $PM_1$ $f$(RH) cycles are then broken down by wind direction quadrants (northeast (NE), southeast (SE), southwest (SW), and northwest (NW)) on a **(c)** monthly and **(d)** hourly basis to show the changes in source regions. The median
$f$(RH) from all of the data is represented as a grey dashed trace in (c) and (d).

3.2.4.  Wind sector analysis of optical properties and $f$(RH)

        To elucidate the impact of different source regions, and subsequent seasonal changes in those sources,

$PM_{10}$ aerosol optical properties were binned both by wind direction and wind speed for summer and winter data

(**Fig. 8**). This wind sector analysis could not be done with the ion mass concentration data because of insufficient

time resolution in the data ($PM_1$ sampled on 24 hr intervals and $PM_{10}$ sampled over week long periods), which



would mask any wind-related information. However, possible changes in composition as a function of wind direction are supported by directional patterns in $f$(RH) explored previously. Only data after October 7, 2003 was used in this analysis in order to ensure consistent treatment of the data and remove periods where wind direction-based contamination screening process was applied (**Section 2.2**).

Significant differences between optical properties in the summer and winter are apparent in terms of magnitude, distribution across wind sectors, and relation to wind speed. Summertime aerosol number concentration and scattering exhibit more variability across wind directions and speeds but the patterns look quite similar to each other suggesting a common source (**Fig. 8a, e**). Both number concentrations and scattering are high when the wind is from the north and relatively strong (~15 m s$^{-1}$) and also when the wind is from the southeast and the wind speed is low (<5 m s$^{-1}$). In the summer, absorption coefficients (**Fig. 8c**) are highest in air coming from the northeast quadrant – this peak in absorption only occurs when wind speeds are low (<5 m s$^{-1}$) suggesting a local source. In contrast, in the winter, number concentration and absorption cycles (**Fig. 8a, c**) look similar to each other and both exhibit the highest amount for low wind conditions (i.e., close to the center of each plot), suggesting more local influence and a common source. The highest wintertime scattering comes from the northwest when wind speeds are high, suggesting increased wind-driven sea spray emissions (**Fig. 8e**).

Calculated optical properties provide more information about the characteristics of the aerosol in the different sectors. In the winter, there is a clear anthropogenic signal in the northeast quadrant. Values of SSA are low while SAE and BFR are both high, suggesting smaller, darker aerosols such as those from combustion sources (**Fig. 8g, i, j**). This is likely related to home heating from wood burning (City of Arcata, 2008). The low wintertime SSA values correspond with low amounts of scattering in the northeast quadrant (scattering values are less than 10 Mm$^{-1}$) and higher amounts of absorption, possibly due to preferential removal of scattering aerosol. Low SAE values also correspond with the lowest observed $f$(RH) (**Fig. 8b**), which is consistent with combustion-related aerosol (Maßling et al., 2003; Titos et al., 2016). In the summer, the scattering and the absorption in the northeast sector both exhibit high values at low windspeeds and the SSA in the quadrant is generally lower than for the other sectors, again suggesting some sort of local anthropogenic source in that direction. Finally, this temporal cycle is also supported by higher SAE values indicating a larger contribution of submicron aerosol. It is important to recall that the winds at THD are relatively rare from the northeast quadrant (~9% overall, ~7% in summer and ~14% in



winter, e.g., **Fig S3**), so the overall contribution from the quadrant to the aerosol climatology is limited. This was

observed previously in the climatology of $f$(RH) values.

    For both summer and winter, SAE values are lower at higher windspeeds (> 10 m s$^{-1}$), consistent with

wind-driven emissions of sea spray (Vaishya et al., 2012). During the summer, the highest SAE values occur during

southwesterly flow. This is potentially related to marine biogenic activity, as has been suggested in studies for other

marine influenced sites (Quinn et al., 2002; Yoon et al., 2007) and which is supported by the $f$(RH) observations for

the southwest quadrant (**Section 3.2.3**). The link between AAE and marine air masses is not clearly defined. For

example, Schmeisser et al. (2017) observed that though many of the marine air masses at the sites they studied

tended towards lower AAE no clear pattern could be characterized. The AAE at THD, although exhibiting different

values over the seasons, shows a shift from higher AAE in the east (land) to lower AAE in the west (ocean) which

generally suggests increased influence from clean marine air (**Fig. 8h**). This is expected given the geography of

THD.

    The relationship between AAE and SAE is thought to be a possible indicator for the type of aerosol being

measured (Cappa et al., 2016; Cazorla et al., 2013), however, these regimes have been shown to be less reliable at

locations with complex conditions and source types (Schmeisser et al., 2017). In this work we used the classification

scheme from Cappa et al. (2016) to look at general regimes; separating AAE and SAE (450 & 700 nm) comparisons

by both wind direction and season along with aerosol size in order to more effectively classify changes in aerosol

types at THD (**Fig. S11**). PM$_1$ winter data was largely concentrated in the BC dominated and BC/ brown carbon

mixture regions, this shifted down towards the small particle/ low absorption region in the spring through fall with

the highest concentration of data in this region during the summer. In all four seasons the data from the western and

eastern quadrants were stratified, with higher ratios for easterly winds (45 – 135° consistently had the highest ratios

of AAE to SAE). Along with the other evidence presented, this supports increased combustion sources in the winter

and fall that mainly come from the east with more biogenic activity in the summer from the west. Comparisons for

the PM$_{10}$ data were similar, although predictably shifted toward larger particle regimes. In contrast to the other

seasons, the summer PM$_{10}$ data from the west remained in the small particle/ low absorption group and didn't shift

into the large particle/ low absorption region. Higher Rsp values in the summer support this and point to the

formation of small biogenic aerosol during the summer to the west of THD (**Fig. 8f**).



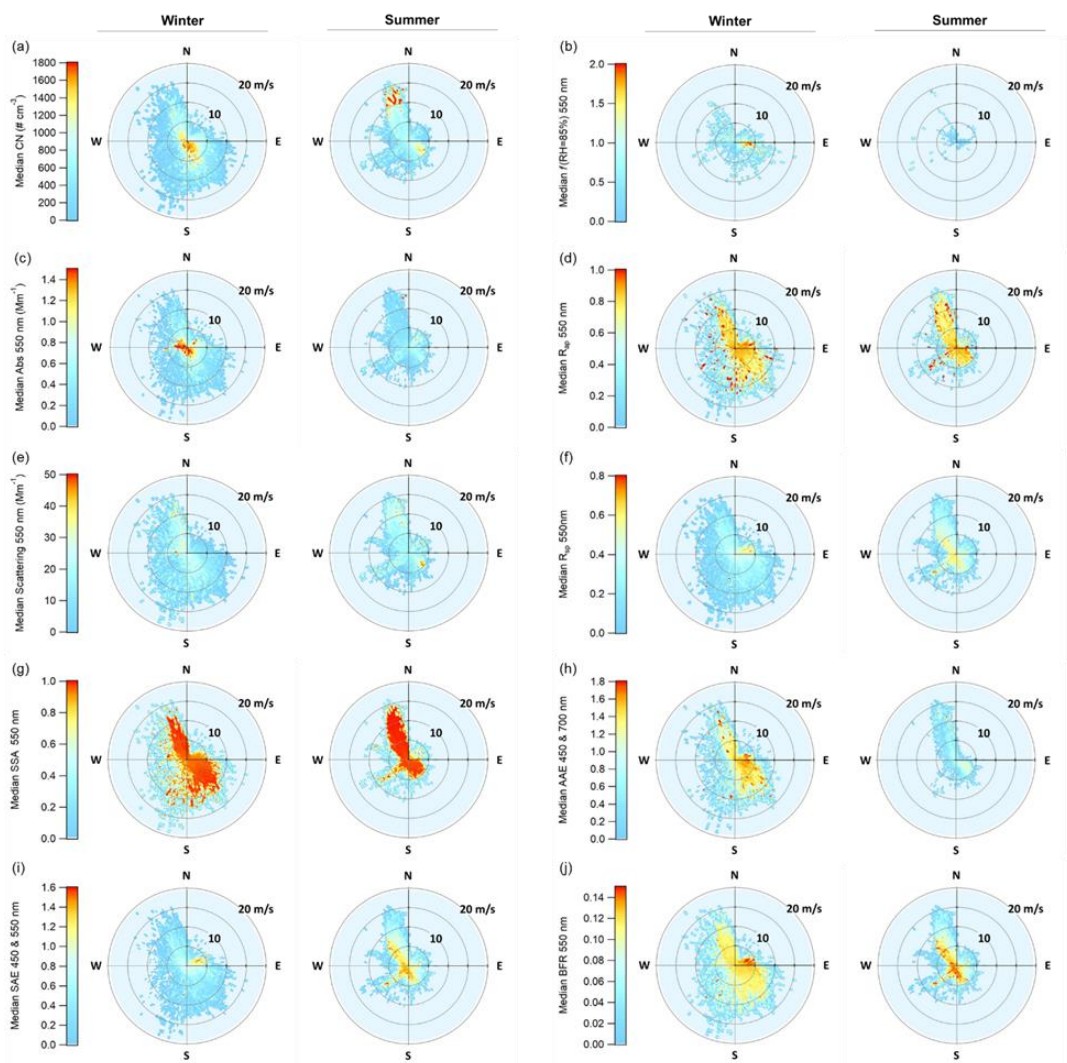

**Figure 8**: Wind vector analysis for PM$_{10}$ aerosol optical properties and $f$(RH=85%), showing the median value at a given wind speed and direction. For each property the winter is on the left and the summer on the right, a color scale on the left applies to both figures.


*3.3. Patterns in aerosol composition and optical properties*

Aerosol chemical and optical properties have been linked previously at other marine sites. For example, aerosol

data from NOAA's Barrow observatory (Utqiaġvik, AK, USA) exhibited a strong ($R^2 = 0.8$) correlation between

MSA$^-$ and CN concentration during the summer (Quinn et al., 2002) which they attributed to particle formation from



biogenic sources. Quinn et al. (2002) also performed linear regressions of aerosol scattering versus derived 'sea salt'

and nss $SO_4^{2-}$ concentrations at Barrow and found relatively strong positive correlations ($R^2 > 0.5$) for some seasons.

At THD a negligible number of significant relationships ($r > 0.5$) are observed between ion and CN concentrations,

however, several relationships between ion chemistry, scattering and absorption exist (**Table S9, S10, S11, and**

**S12**). Scattering coefficients ($\sigma_{sp}$) correlate with $PM_1$ mass in the winter and summer. In the summer $\sigma_{sp}$ has a

significant relationship with MSA in July and in August, but not in June, indicating biogenic production of small

scattering aerosols in those months. Interestingly, $\sigma_{sp}$ correlates with both K and oxalate in the winter ($\sigma_{sp}$ and $SO_4^{2-}$

relationships are significant in December and close to significant in January) and the absorption coefficient ($\sigma_{ap}$)

correlates with $SO_4^{2-}$, K, and oxalate only in the January data. These individual ion to optical relationships are

consistent with the PMF analysis discussed previously (**Section 3.1.4**). The $PM_1$ anthropogenic/ combustion factor

correlates with the absorption coefficient, indicating that it is driving the increased absorption in the colder months

(**Fig. S12**). While neither the sea salt nor biogenic factors had significant relationships with the scattering

coefficient, the biogenic patterns were the stronger (larger $r$) of the two in both seasons indicating a stronger impact

on scattering (**Fig. S12**).

Another approach to looking at relationships between aerosol chemistry and optical properties is to segment the

data by SSA (550 nm) and SAE (450 & 550 nm) and look for differences in mass concentrations for the different

groupings of data. This segmentation was done for the $PM_1$ data (**Fig. 9**), using the 25th and 75th percentiles of SSA

(0.89 and 0.95) and SAE (1.91 and 2.24). Five aerosol groups were defined using the following bounds: big-

scattering (SSA > 0.95, SAE < 1.91), small-scattering (SSA > 0.95, SAE > 2.24), mixed (0.89 < SSA < 0.95, 1.91 <

SAE < 2.25), small-absorbing (SSA < 0.89, SAE > 2.24), and big-absorbing (SSA < 0.89, SAE < 1.91). These group

classifications are linked to common aerosol types, such as, marine (big-scattering), secondary nucleation (small-

scattering), anthropogenic (small-absorbing), and dust (big-absorbing). Bigger, more scattering aerosols had higher

concentrations associated with calculated sea salt and its associated ions ($Cl^-$, $Na^+$, and $Mg^{2+}$) while smaller, more

scattering aerosols had higher concentrations of biogenic related ions ($MSA^-$, $SO_4^{2-}$ (total and nss fraction), and

$NH_4^+$). For darker aerosols, the median for the small-absorbing mass concentration was larger than that of the big-

absorbing mass concentration for every species investigated except $Br^-$. Looking at the mass for each of these groups

within a given species, patterns in source related ions and chemical components can be observed (**Fig. 9, S13**).



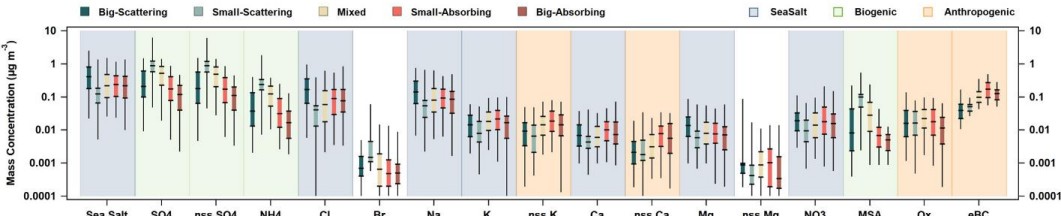

**Figure 9**: PM$_1$ ion mass concentrations, as well as sea salt and M$_{eBC}$, as a function of SSA (550 nm) and SAE (450 & 550 nm). For each species mass of big-scattering (SSA > 0.95, SAE < 1.91), small-scattering (SSA > 0.95, SAE > 2.24), mixed (0.89 < SSA < 0.95, 1.91 < SAE < 2.25), small-absorbing (SSA < 0.89, SAE > 2.24), and big-absorbing (SSA < 0.89, SAE < 1.91) aerosol fractions are listed from left to right and color coded as outlined in the top left legend. Additionally, the likely source of the ion is indicated using shading. Blue shading indicates a sea salt related component, green indicates biogenic components, and orange indicates anthropogenic/ combustion relationships (in right top legend).

### 3.4. THD as a representative marine site

THD was part of the NOAA Federated Aerosol Network (NFAN) which also includes several clean marine sites: Cape Grim, Australia; Cape Point, South Africa; Samoa, American Samoa; Cape San Juan, Puerto Rico; and Sable Island, Canada (Andrews et al., 2019). Data from these sites compared to that from polluted marine environments shows a strong separation between aerosol optical properties (**Table 3**). Clean marine sites tend to meet the following criteria: $\sigma_{sp} < 40$ Mm$^{-1}$, $\sigma_{ap} < 1$ Mm$^{-1}$, SSA > 0.97, and SAE < 1. Anthropogenically impacted sites fail to meet one or more of these constraints. Diurnal variability at clean marine sites is minor, as shown by Delene and Ogren (2002) who reported virtually no diurnal variability in either SSA or BFR at Sable Island (1992 – 2000) suggesting minimal influence from local sources. THD has diurnal cycles suggestive of anthropogenic influences, similar to those observed by Bhugwant et al. (2001) for absorbing aerosol at a coastal site on La Reunion Island. However, looking at THD data as a whole it aligns most closely with other clean marine sites (**Table 3**). Comparison to vertical profiles of absorption over the Pacific during the multi-year ATOM (Atmospheric Tomography Mission) campaign can also clarify THD's status as a clean marine site. During ATOM the average mass mixing ratio of refractive black carbon (rBC) in the 20-60 N latitude over the Pacific Ocean from ATOM was approximately 10 ng/kg at the lowest flight level (~0.2 km), which is equivalent to an $\sigma_{ap}$ of ~1.2 Mm$^{-1}$ (assuming dry air mass density of 1.2 kg m$^3$ and MAC 10 m$^2$ g) (Katich et al., 2018). This is well within the range of the aerosol $\sigma_{ap}$ measured at THD (**Fig. 5, 6**), with an annual median $\sigma_{ap}$ of 0.62 Mm$^{-1}$. This leads to the conclusion that,




even with influences from local vehicular and marine traffic, seasonal wood burning, and marine biogenic

emissions, the majority of measurements made at THD are representative of clean marine air.

**Table 3**: In situ annual aerosol property statistics at long-term coastal monitoring sites (PM$_{10}$ or no size cut). Values reported at green wavelength (at or near 550 nm) for scattering, absorption, and SSA unless otherwise noted. Presented SAE values were calculated for a **blue/green** wavelength pair unless otherwise indicated (*). Values are medians unless only available statistic was mean value. Shorter term studies of these properties are also listed at the bottom of the table.


| | Site | Scattering (Mm$^{-1}$) | Absorption (Mm$^{-1}$) | SSA | SAE |
|---|---|---|---|---|---|
| **Clean Marine** | THD | 21.5 | 0.62 | 0.97 | 0.78 |
| | Cape Point, South Africa[1] | 18.8 | 0.26 | 0.98 | 0.43 |
| | Samoa, American Samoa[1] | 21.4 | - | - | -0.16 |
| | Mace Head, Ireland[2, 3, 4] | 10 - 30 | 0.15 – 0.20 | 0.94 – 1.0 | 0.2 – 0.8* |
| | Cape San Juan, Puerto Rico[1] | 29.0 | 0.75 | 0.97 | 0.35 |
| | Sable Island, Nova Scotia[1] | 34.0 | 0.66 | 0.98 | 0.75 |
| | **Average** | **24 ± 8** | **0.4 ± 0.3** | **0.97 ± 0.02** | **0.4 ± 0.4** |
| **Anthropogenically Influenced Marine** | El Arensosillo, Spain[1] | 35.4 | 3.23 | 0.91 | 1.38 |
| | Gulf of Cadiz, Spain[6] | 38.0 | - | - | 1.47* |
| | Finokalia, Greece[5] | 50 | 5.6 | 0.89 | - |
| | Gosan, South Korea[1] | 72.8 | 5.60 | 0.93 | 0.75 |
| | Hok Tsui, Hong Kong[7] | 134 | 6.6 | 0.94 | 1.4* |
| | Anmyeon-do, South Korea[1] | 113.9 | 9.69 | 0.92 | 1.48 |
| | **Average** | **70 ± 40** | **6 ± 2** | **0.92 ± 0.02** | **1.2 ± 0.4** |

* SAE was measured for the blue and red wavelengths
[1]Andrews et al. (2019), [2]Vaishya et al. (2011), [3]O'Dowd et al. (2012), [4]Jennings et al. (2003),
[5]Vrekoussis et al. (2005), [6]López et al. (2015), [7]Wang et al. (2017)



## 4.  Conclusions: Aerosol sources at THD and THD as a representative marine site

This paper presents the seasonal climatology of aerosol chemical and optical properties from a 15-year data set obtained at NOAA's now closed Trinidad Head observatory. As expected, sea salt aerosol is a consistent source at THD and was identified as a source factor in every season (**Fig. 4**). Calculated sea salt mass, and mass of its associated ions, consistently dominates $PM_{10}$ and contributes strongly to $PM_1$ in the winter (**Fig. 1, 2**). A consistent $PM_{10}$ SAE of less than 1.5 and Rsp of less than 0.6 also show the influence of large non-absorbing sea salt

throughout the year (**Fig. 5**). Over all seasons, SAE was lower and concentrations of sea salt ions increased at higher windspeeds, indicating wind-driven sea spray emissions that mainly come from the western (ocean-side) quadrants around THD (**Fig. 8**). Additionally, aerosols at THD are largely composed of acidic compounds as demonstrated by the relationship between sulfate and ammonium ions in all seasons.

        Aerosol ion chemistry and optical properties at THD exhibit monthly and diurnal temporal cycles, suggesting

changes in sources, sinks, and transport throughout the year. Biogenic activity contributes to aerosol concentration throughout the year, however, the source is much stronger in the summer (**Fig. 1, 2, 4, S4**). Large contributions of these small scattering aerosols contribute to both $PM_1$ and $PM_{10}$ the summer (**Fig. 5, 9, S13**). While daily cycles in aerosol optical properties indicate some influence from local traffic emissions (**Fig. 6**), the winter is the only season with a persistent anthropogenic/ combustion source (**Fig. 7, 8**). This source is only identifiable in the winter (**Fig. 4,**

**S4**) – likely from wood burning home heating – and is the driver of increased absorption during this time (**Fig. 5, S12**). These sources could be further parsed into subcategories; however, this would require higher resolution chemical data.




## Data Availability

Data used in this work is available online through several online databases. Aerosol ion data can be accessed through the PMEL Atmospheric Chemistry Data Server (https://saga.pmel.noaa.gov/data/). Aerosol optical data and meteorology data is available through the NOAA Global Monitoring Laboratory database (https://gml.noaa.gov/dv/data/). The $f$(RH) data are available through the ACTRiS Data Center (https://actris.nilu.no/Content/?pageid=deba50b668fc4ce6b98afbc97bdc4025).

## Author Contribution

EKB preformed the final analysis presented in this work and prepared the manuscript with contributions from all co-authors. EA did initial analysis for this work, wrote significant portions of the manuscript, and guided the research. EA, PJS, and PKQ collected the data used in this work. EA and PJS corrected and quality controlled the aerosol optical data and PKQ analyzed the filter samples for the aerosol ion data. All authors reviewed and edited the
manuscript.

## Competing Interests

The contact author has declared that none of the authors has any competing interests.

## Acknowledgments

This research was supported by NOAA cooperative agreements NA17OAR4320101 and NA22OAR4320151 for E.B. and E.A. This is PMEL contribution number 5446 for P.K.Q. We thank Michael Ives, Wendy Snible, and other THD station personnel. Special thanks to Jim Wendell for engineering support and Derek Hageman for help with data acquisition, processing, and archiving of meteorological and aerosol data. We thank Steven Cliff, Kevin Perry, and Yonjing Zhao for analyzing the DRUM data shown in the supplemental – funded by NOAA award
NA16GP2360 and supported by the Department of Energy, Office of Basic Energy Science.

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
