# Peer review of "Climatology of aerosol properties at an atmospheric monitoring site on the Northern California coast"

_EGUsphere, 2023_

## Author Response (AR1)

**Author response to reviews**

We would like to thank both reviewers for their time and consideration of this manuscript. The comments from both reviewers are addressed here, with the original comment italicized and our response bellow.

*I recommend adding a map that includes major cities such as San Francisco, Eugene, and Arcata.*

*A map indicating the site and neighboring locations would be helpful.*

- Both reviewers requested that a map be added to give context to the manuscript. We appreciate this comment and have prepared the following figure to show the position of the measurement site in relation to surrounding cities:

[Figure]

**Figure 1**: Maps of the measurement site in relation to major cities. A smaller scale is shown in (a) of the more local coastline, and a larger scale in (b) shows the broader area. Maps were constructed using images and information from the United States Geological Survey's program The National Map (US Geological Survey, 2019).

Reference to The National Map has also been added:

U.S Geological Survey: The National Map - New data delivery homepage, advanced viewer, lidar visualization, U.S. Geological Survey Fact Sheet 2019-3032, 2pp., https://doi.org/10.3133/fs20193032, 2019.

And in Line 77 we have added the following information:

"The site is ~22 km (~13.5 mi) North of the smaller city of Arcata (population ~19,000) and directly Southwest of the city of Trinidad (population ~300)."

*Line 199: Remove the second "that"*

- In line 199, the second that has been removed.

*Given that the chemical analysis took place during a 4-year period, were there any trends observed for different components? For example, Weber et al. (2016) observed a declining trend in sulfate and ammonium at the southeastern United Stated during the past 15 years.*

- We'd like to first note, that the 4 years of chemical ion data is not long enough to identify or define a trend in aerosol composition at THD. To do that we'd need at least 10 years of data (Weatherhead et al., 1998).

In terms of general changes, we saw no statistically significant changes in the PM1 or PM10 ion data from 2002 – 2006. Here are the timeseries of monthly median values for all of the PM1 and PM10 main ion components. Note that all slopes are near zero, and none of the fits are significant. Given that trend analysis is outside the ability of this dataset, these will not be included in the manuscript.

[Figure]

*Some additional information concerning the PMF analysis could be helpful; what was used as an independent variable? PM1 mass? Or just the sum of the derived ion composition analysis?*

-The independent variable was the sum of the identified ions. Because the unknown/ unidentified fraction of the filter masses were so large (60-80%), using the total PM1 mass could have overestimated the factor contributions. This sum was well predicted by the PMF model in both the summer (slope = 1.05; int. = -0.07; r2 = 0.986) and winter (slope = 0.966; int. = -0.02; r2 = 0.931), which further showed that it was appropriate to use. The error assigned to the total mass in the PMF model was propagated from the errors of the individual ions.

We have added the following to the methods section (Line 226):

"Since there was a large fraction of unidentified mass in the filter samples (**Section 3.1.1**), the sum of the identified ion mass and not PM1 was used as the independent variable and total mass in the PMF model in order to avoid overestimating the factor contributions.

And to the results (Line 411):

"Good correlations were observed between the predicted and observed total identified ion mass for the winter (y = 0.966x -0.02, r2 = 0.931, Fig. S7a) and the summer (y = 1.05x -0.07, r2 = 0.986, Fig. S7b)."

Additionally, Figure S7 was updated to show these comparisons along with the factor contribution and mass:

[Figure]

**Figure S7**: Predicted mass vs observed mass for the (a) winter and (b) summer. Timeseries of factor (c) contribution (normalized so that the average contribution in each season is 1) and (d) mass for the winter and summer PMF analysis.

*Why do authors consider that all non-sea salt sulfate is of biogenic origin? Can't it also be from secondary formation of anthropogenic origin? For example, Kirpes et al. (2018) find internally mixed secondary sulfate with sea spray aerosol in the marine environment of the Arctic during winter.*

- For the non-sea salt sulfate attribution, our intention was not to attribute it all to biogenic origin. Our apologies if this is how the manuscript is currently reading.

Nearly all of the PM1 sulfate is nss in origin (Fig. 3), and it's likely that the site is getting anthropogenic sulfate especially during colder months when there is increased influence from anthropogenic/ combustion emissions. The oxalate to nss sulfate ratio supports this, and the fact that sulfate is always significantly correlated with oxalate indicates that it is likely coming from both sources just as oxalate is. The PMF analysis also supports this, attributing a non-trivial portion of the sulfate signal to the anthropogenic/ combustion factor in the winter. The SSA and SAE aerosol type characterization was not meant to define a single source for each ion. The sea salt/ biogenic / anthropogenic highlights in Fig. 9 were meant to show the patterns in the distribution across the aerosol types with regards to a possible dominate source for ions with those patterns, but it was not meant to rule out the other sources as contributors.

In the Fig. 9 caption we have rephrased to indicate that these shading indicates related components, but not absolute sources: "Additionally, one possible source of the ion based on its distribution across the aerosol types is indicated using shading. Blue shading indicates a sea salt related component, green indicates biogenic related components, and orange indicates anthropogenic/ combustion relationships (in right top legend)."